# CAMoE: Cost-Aware Communication Optimization for Mixture-of-Experts Inference

## Abstract

Mixture-of-Experts (MoE) is currently the most promising method for scaling the parameters of large language models. Its architecture consists of different experts at different layers, with a fixed number of top experts selected dynamically for each token based on the token's information during inference. Ideally, if all experts could be placed on the same device, token routing would not be impeded by communication overhead. However, as the parameters of MoE models grow toward trillion-scale, experts cannot be accommodated on a single device or even a single node, leading to significantly increased tail latency during all-to-all communications—the tokens with the highest communication cost slow down the inference process.

In this paper, we thoroughly analyze the patterns of all-to-all communications during inference in MoE models and develop a profiler to measure heterogeneity between devices. Using parameters obtained from profiler runs, we implement a SystemC-based simulator to model communication times during all-to-all communications. Based on detailed information about transmitted data, we propose a cost-aware method designed to reduce tail latency during model inference. Experimental results demonstrate that this method does not affect model accuracy on downstream tasks and effectively reduces all-to-all communication time during inference. Our implementation is publicly available at https://anonymous.4open.science/r/CAMoe-1FBB.

## 1 Introduction

Large Language Models (LLMs)(OpenAI et al., 2024)(Touvron et al., 2023)(DeepSeek-AI et al., 2025)(Yang et al., 2025) have advanced rapidly in both capability and scale, demonstrating strong performance in reasoning, coding, and multi-turn interaction. This progress is tightly coupled with growth in parameter counts and compute investment, and with architectural evolution from dense Transformers to sparse or modular forms. While larger models often increase utility, they also strain training and inference budgets: memory footprints grow with parameters, and end-to-end latency/throughput hinge on how efficiently the model exploits parallel hardware at deployment.

Mixture-of-Experts (MoE) architectures expand total parameter capacity while keeping per-token FLOPs nearly constant by activating only a *sparse* subset of experts per token. This conditional computation yields a favorable scaling path: increases in capacity primarily drive memory and *communication* overhead rather than per-token compute. Consequently, MoE has become a leading approach for scaling LLM capacity in production environments.

Early MoE introduced auxiliary balancing losses to achieve more uniform token-to-expert assignment and stabilize training (Shazeer et al., 2017; Lepikhin et al., 2020); Switch Transformer(Fedus et al., 2022) simplified routing to top-1 with capacity control to reduce FLOPs while preserving quality. More recent methods adjust the number of active experts per token or alter routing dynamics (e.g., XMoE, Adaptive Gating) to trade FLOPs for quality (Yang et al., 2024; Li et al., 2023). These approaches primarily optimize *model-side* metrics (accuracy, FLOPs, overflow) and typically assume a homogeneous communication environment, so they do not directly target *system-level* All-to-All (A2A) latency on heterogeneous hardware.

We propose **CAMoE**, a training-free method that injects a small, deterministic, topology- and traffic-aware bias into the router's logits. The bias is a row-wise $z$-scored estimate of communication

time from each source endpoint (where a token resides) to each expert endpoint, computed via a lightweight two-phase profiler and a direction-specific $\alpha$–$\beta$ model. At runtime, CAMOE reduces to a single gather-and-add before the usual Top-$K$ gating: no retraining, no new collectives, no changes to capacity control, and full compatibility with data, tensor, pipeline, and expert parallelism.

**Comparison with prior work.** CAMOE distinguishes itself from prior work in several key ways. Unlike gating-only methods (Shazeer et al., 2017; Lepikhin et al., 2020; Fedus et al., 2022; Yang et al., 2024; Li et al., 2023), which primarily focus on FLOPs or token-count balance, CAMOE is not a new router objective but rather augments existing gating with a normalized cost term to explicitly target A2A mean and p95 latency. In contrast to topology-aware training and placement strategies (e.g., TA-MoE) (Chen et al., 2023), it requires neither offline re-placement nor retraining; instead, it operates under any fixed expert layout by steering tokens away from high-cost paths, making it complementary to improved placements. Furthermore, when compared with systems and engineering stacks (DeepSpeed-MoE, FasterMoE, etc.) (Rajbhandari et al., 2022)(He et al., 2021), CAMOE functions as a router-side drop-in that preserves underlying kernel and overlap optimizations, as it acts at a different layer of the stack. It also achieves communication awareness without the complexity of inference-time deployment and scheduling (Huang et al., 2024), relying on a single, stable hyperparameter $\lambda_{\text{cost}}$ thanks to its row-wise normalization.

Finally, CAMOE is orthogonal to inter-layer affinity and regularization techniques (Yao et al., 2024; Muzio et al., 2024); while affinity reduces cross-layer routing churn, CAMOE reduces per-layer path cost, allowing the two approaches to be combined for additive benefits.

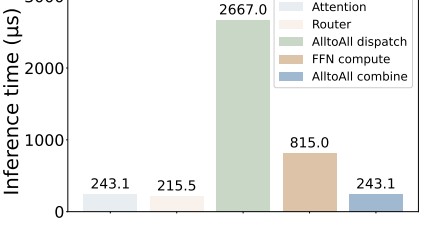

Figure 1: Breakdown of MoE inference components, highlighting the dominance of All-to-All dispatch time.

We summarize the paper's contributions as follows:

**A drop-in, inference-time, topology- and traffic-aware router bias.** A small, deterministic logit bias computed per (source endpoint, expert endpoint) from a lightweight profiler and a direction-specific $\alpha$–$\beta$ link model. It is row-wise $z$-scored, governed by a single hyperparameter $\lambda_{\text{cost}}$, requires no retraining or new collectives, and composes with data/tensor/pipeline/expert parallelism.

**An open-sourced lightweight cost-aware toolkit.** We provide a minimal profiling/simulation toolkit that fits $(\alpha, \beta)$ per direction, exports per-endpoint cost tables, and plugs into Megatron-LM with a concise gather-and-add integration for cost-aware routing.

**End-to-end performance gains with negligible accuracy movement.** On Qwen3-30B-A3B, the method reduces per-layer All-to-All mean latency by up to 15.8% and p95 by up to 19.1%, with small average accuracy changes across nine downstream tasks; a Pareto region $\lambda_{\text{cost}} \in [0.15, 0.20]$ offers strong latency gains at near-baseline quality (cf. Table 1).

## 2 BACKGROUND AND MOTIVATION

### 2.1 MIXTURE-OF-EXPERTS (MOE)

Mixture-of-Experts (MoE) models scale neural network capacity via conditional computation: a learned router activates only a sparse subset of specialized FFN experts for each token, thereby expanding total parameters while keeping per-token FLOPs nearly constant. As illustrated in Figure 2(a), an MoE layer typically replaces the conventional feed-forward network within a Transformer decoder block with a bank of experts governed by the router.

The selection of experts is governed by a learned gating network (**router**), which determines the most relevant experts for processing each token. A commonly used choice in prior work is the *Noisy Top-k Gating* mechanism. For completeness, we briefly summarize its formulation. Given an input representation $x \in \mathbb{R}^d$, gating weights $W_{\text{gate}} \in \mathbb{R}^{d \times N}$, noise weights $W_{\text{noise}} \in \mathbb{R}^{d \times N}$, and a set of

$N$ experts $E_i i = 1^N$, the router first computes the base logits and applies noise:

$$H(x) = x \, W_{\text{gate}} \tag{1}$$

$$\sigma(x) = \text{Softplus}(x \, W_{\text{noise}}) \tag{2}$$

$$\tilde{H}_i(x) = H_i(x) + \sigma_i(x)\, \xi_i, \quad \xi_i \sim \mathcal{N}(0,1) \tag{3}$$

Next, to enforce sparsity, only the top $k$ experts are selected. Let $T$ be the set of indices corresponding to the top $k$ values in the noisy logits $\tilde{H}(x)$. The router then creates masked logits $\hat{H}(x)$ where non-selected expert logits are set to $-\infty$:

$$\hat{H}_i(x) = \begin{cases} \tilde{H}_i(x), & \text{if } i \in T \\ -\infty, & \text{otherwise} \end{cases} \tag{4}$$

The final gating scores are obtained via a softmax over the masked logits:

$$G(x) = \text{softmax}(\hat{H}(x)) \tag{5}$$

Finally, the MoE layer aggregates the outputs from all experts, weighted by their gating scores. Since the scores for non-selected experts are zero, this effectively combines the outputs of only the top $k$ experts:

$$\text{MoE}(x) = \sum_{i=1}^{N} G_i(x)\, E_i(x) \tag{6}$$

Within Transformer-based architectures, MoE layers are integrated into the decoder blocks by replacing conventional Feed-Forward Networks (FFN). Each decoder block comprises a layer normalization step, followed by masked self-attention, and subsequently, an MoE sub-layer that selectively routes input tokens to different experts based on the router's decision. After expert processing, the outputs from different experts are combined to maintain the integrity of the token sequence and fed into subsequent layers.

Although MoE architectures reduce per-token computation via sparse expert activation, the total parameter count (and thus model/optimizer states) still grows with the number of experts, making it infeasible to replicate all experts on every device. *Expert parallelism* addresses this by sharding the expert bank across multiple GPUs or compute nodes—while typically replicating the non-expert layers—as depicted in Figure 2(b). This design (i) unlocks capacities far beyond a single GPU's memory budget, (ii) improves throughput by aggregating many token assignments into large, well-batched GEMMs on each device at nearly constant per-token FLOPs, and (iii) composes cleanly with data/tensor/pipeline parallelism for multi-dimensional scaling. Because tokens must be processed by the specific experts they select, EP necessarily introduces communication to route tokens to the owning devices and reassemble the sequence; operationally this is realized with two collective operations:

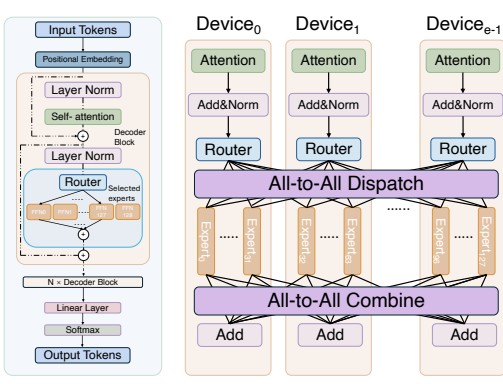

(a) Structure of an Transformer decoder block with MoE.

(b) Communication in expert parallelism for MoE models.

Figure 2: Illustration of MoE architecture and parallel communication strategy.

**All-to-All Dispatch**: Redistributes input tokens across different devices according to their assigned experts, ensuring that each token reaches the device hosting its corresponding expert.

**All-to-All Combine**: Collects processed token outputs from all devices, restoring the original token order on the originating devices.

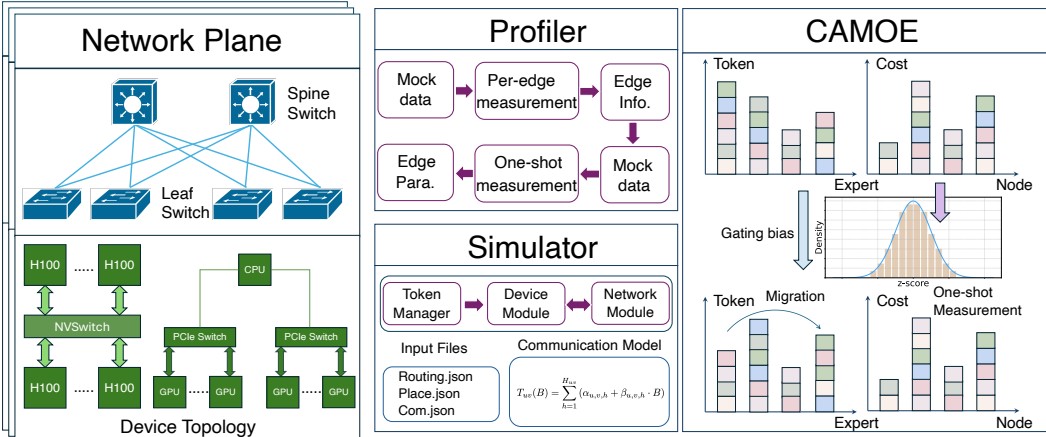

Figure 4: **Real-deployment view and CAMOE pipeline.** Left: *Network Plane* (spine–leaf fabric) and *Device Topology* (NVSwitch/NVLink/PCIe Switch/host paths). Middle: *Profiler* generates per-edge parameters and a one-shot calibration; *Simulator* predicts All-to-All time from routing/place-ment and a hop-wise communication model. Right: *CAMOE* injects a deterministic, topology-aware cost bias into gating, normalizes costs (z-score), and migrates a small fraction of tokens off expen-sive links while preserving accuracy.

## 2.2 MoE INFERENCE IN REAL DEPLOYMENT

As illustrated in Figure 1, the All-to-All dispatch step overwhelmingly dominates inference latency, significantly surpassing other computational components such as attention mechanisms, routing pro-cesses, and Feed-Forward Network (FFN) computations. More importantly, in real deployment environments, node resources and network connectivity are typically heterogeneous and variable, making these production environments particularly susceptible to tail latency issues. This inherent heterogeneity further exacerbates the unpredictability and inefficiency of actual inference perfor-mance. Therefore, in this section, we provide a concise overview of heterogeneous connectivity within such real-world deployment environments and discuss the resulting tail latency phenomena.

### 2.2.1 HETEROGENEOUS CONNECTIVITY IN REAL DEPLOYMENTS

Large production clusters commonly combine NVLink/NVSwitch, PCIe, and host paths, and are operated under practical constraints (scheduler fragmen-tation, partial GPU availability on nodes, maintenance windows, and cross-rack placement). These factors create diverse communication costs that directly impact MoE collectives.

**Intra-node** However, a critical and often-overlooked challenge in the real-world deployment of modern multi-GPU servers is the extreme heterogeneity of intra-node communication. While it is commonly assumed that inter-GPU communication within the same server is ho-mogeneous and efficient, our empirical measurements re-veal a starkly different reality.

As shown in Figure3, depending on the physical inter-connect topology between a pair of GPUs, their point-to-point write latency can differ by orders of magnitude. Communication can be routed over high-speed direct

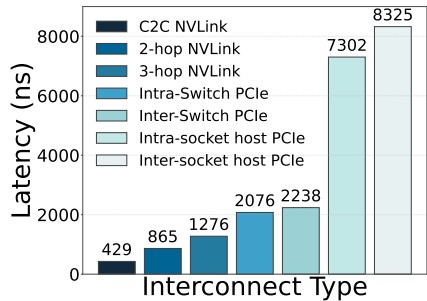

Figure 3: Intra-node GPU-to-GPU write-operation latency under different interconnect types.

NVLink (latency 429 ns), multi-hop NVLink requiring data forwarding, next-best PCIe switches,

or in the worst case, host paths that detour through the CPU (latency as high as 8325 ns). The performance gap between the fastest and slowest paths is nearly a factor of 20.

**Inter-node** Beyond a single server, production clusters are built as spine-leaf network fabrics, as illustrated on the Network Plane of Figure 4. Due to the static and exclusive allocation mechanism of GPU resources, resource utilization within nodes is often insufficient, forcing some tasks to be distributed across multiple nodes. When the resources of a single node cannot fully meet the requirements of an individual task, the task must be allocated across multiple nodes for execution(Wu et al., 2023)(Amaral et al., 2017)(Xiao et al., 2018).

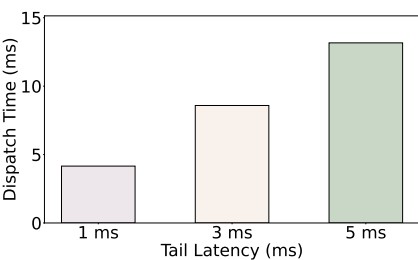

All-to-All is a barrier-synchronized collective communication: a step completes only when every destination node has received all of its messages. Therefore, a single slow receiver—whether caused by a high-cost path or transient congestion—gates the completion of the entire step. As illustrated in figure, we use Linux's `tc` utility to perturb a single device's path, injecting additional latency while holding other devices near their baseline.

The measured All-to-All dispatch time tracks the slowest node almost one-for-one: as the injected delay grows, $T_{A2A}$ increases nearly linearly with the maximum per-node latency, while reducing delay on non-bottleneck nodes has a negligible effect on the overall step time.

Figure 5: Impact of tail latency on the dispatch time in All-to-All communication, indicating that the slowest node significantly determines overall performance.

### 2.3 MOTIVATION

A2A is fundamentally a barrier-synchronized collective communication: the entire step completes only when its slowest receiver has finished. Our perturbation experiments corroborate this; increasing delay on a single receiver node inflates the total time of $T_{A2A}$ almost one-for-one, whereas accelerating non-bottleneck nodes has a negligible effect. This mechanism is a direct driver of tail latency.

Furthermore, router behavior is another, more subtle driver of tail latency. At inference time, to ensure determinism, the router's decisions are completely decoupled from underlying link costs. Consequently, a large volume of tokens may be routed to experts that happen to reside behind high-cost communication paths. This not only increases the average data transfer cost but, more critically, exacerbates the load imbalance between nodes, thereby elevating the overall latency ceiling.

Taken together, the core motivation of this work is to address the following question: *How can we effectively mitigate the communication-dominated tail latency of MoE inference in heterogeneous and dynamically changing deployment environments, without sacrificing model accuracy, batching efficiency, or compatibility with existing parallelism strategies?*

## 3 CAMOE

### 3.1 COMMUNICATION PERFORMANCE MODEL

In this section, we formally characterize the communication cost of the All-to-All operations within a single Mixture-of-Experts (MoE) layer. We adopt the classical $\alpha$-$\beta$ model to describe the network fabric's performance, decomposing the communication path between any device pair $(u, v)$ into multiple hops (e.g., NIC-to-ToR, ToR-to-Spine, NVLink, PCIe). Each hop $h$ has a startup latency $\alpha_{u,v,h}$ and a per-byte transmission cost $\beta_{u,v,h}$. Thus, the total communication time for transmitting a payload of size $B$ bytes is expressed as:

$$T_{uv}(B) = \sum_{h=1}^{H_{uv}} (\alpha_{u,v,h} + \beta_{u,v,h} \cdot B) \tag{7}$$

These parameters $(\alpha_{uv}, \beta_{uv})$ can be empirically measured for a given hardware topology.

### 3.1.1 ROUTING-INDUCED COMMUNICATION VOLUME

The data transmission volume between devices directly depends on the gating network's routing decisions. Let $\mathcal{D}$ denote the set of devices, define $\pi(e)$ as the mapping from expert $e$ to its device, and $\sigma(i)$ as the mapping from token $i$ to its device. When token $i$ is routed to $K$ experts, let $C_{ie}$ represent the assignment count from token $i$ to expert $e$. Thus, the number of tokens dispatched from device $u$ to device $v$ is given by:

$$N_{uv}^{\text{disp}} = \sum_{\substack{i:\sigma(i)=u,\\ e:\pi(e)=v}} C_{ie} \tag{8}$$

Assuming no tokens are dropped, the communication volume in the combine step matches the dispatch step, i.e., $N_{uv}^{\text{comb}} = N_{uv}^{\text{disp}}$.

### 3.2 ALL-TO-ALL COMMUNICATION SEQUENCE AND TIMING

Expert-parallel communication within an MoE layer involves three sequential collective operations:

**Preprocess (metadata)**: Expert metadata (token counts per expert), total size $E \cdot s_{cnt}$ bytes.

**Dispatch (embeddings & routing probabilities)**: Token embeddings combined with routing probabilities, total size $(M \cdot s_h + s_p)$ bytes per token.

**Combine (processed embeddings)**: Processed token embeddings returned from experts, total size $M \cdot s_h$ bytes per token.

By explicitly integrating the $\alpha$-$\beta$ model, the total All-to-All communication time within a single MoE layer is formally expressed as:

$$
\begin{aligned}
T_{\text{A2A}} &= T_{\text{preprocess}} + T_{\text{dispatch}} + T_{\text{combine}} \\
&= \underbrace{\max_{(u,v)\in\mathcal{D}^2} \left[ \alpha_{uv} + \beta_{uv} \cdot (E \cdot s_{cnt}) \right]}_{} \text{preprocess (metadata)} \\
&\quad + \underbrace{\max (u,v) \in \mathcal{D}^2 \left[ \alpha_{uv} + \beta_{uv} \cdot N_{uv}^{\text{disp}}(M \cdot s_h + s_p) \right]}_{} \text{dispatch (embeddings \& routing probs.)} \\
&\quad + \underbrace{\max (u,v) \in \mathcal{D}^2 \left[ \alpha_{uv} + \beta_{uv} \cdot N_{uv}^{\text{comb}}(M \cdot s_h) \right]}_{\text{combine (processed embeddings)}}
\end{aligned}
\tag{9}
$$

### 3.3 PROFILER

We estimate $(\alpha_{uv}^p, \beta_{uv}^p)$ for use in equation 9 via a *two-phase, minimal* procedure with no warm-ups or repeated trials: (1) **Baseline** — for each $(u, v)$ and a set of message sizes, measure *isolated* point-to-point transfers and fit $(\alpha_{uv}^{0,p}, \beta_{uv}^{0,p})$; (2) **Congestion-aware** — for each source $u$, issue *one-to-many* patterns, use the baseline model to predict the bottleneck edge, and sample only that edge to refine $(\alpha_{uv}^p, \beta_{uv}^p)$ under contention. The specific algorithm is shown in Appendix1.

### 3.4 METHODOLOGY

**Per-layer expert placement.** For each MoE layer $\ell$, fix the expert→device map $\pi_\ell(e) \in \mathcal{D}$ and the device→endpoint map $\delta : \mathcal{D} \to \mathcal{N}$. Let $\nu_\ell(e) = \delta(\pi_\ell(e))$ be the endpoint of expert $e$. Encode the placement by a one-hot matrix

$$\mathbf{M}_\ell \in \{0,1\}^{E \times |\mathcal{N}|}, \qquad [\mathbf{M}_\ell]_{e,v} = \mathbf{1}\{\nu_\ell(e) = v\},$$

so that $\mathbf{M}_\ell^\top \in \{0,1\}^{|\mathcal{N}| \times E}$ maps endpoint indices to expert indices.

**One-shot traffic accounting.** Run one forward pass with $\lambda_{\text{cost}} = 0$ and stack tokens $i = 1, \ldots, I$. Let the source-endpoint indicator be

$$\mathbf{S} \in \{0,1\}^{I \times |\mathcal{N}|}, \qquad [\mathbf{S}]_{i,u} = \mathbf{1}\{\sigma(i) = u\},$$

and the (pre-drop) routing assignment be

$$\mathbf{A}_\ell \in \mathbb{N}^{I \times E}, \qquad [\mathbf{A}_\ell]_{i,e} = C_{ie}^{(\ell)}.$$

Then the dispatched token-count matrix (source $u$ to expert-endpoint $v$) is

$$\mathbf{N}_\ell^{\mathrm{disp}} = \mathbf{S}^\top \mathbf{A}_\ell \mathbf{M}_\ell \in \mathbb{N}^{|\mathcal{N}| \times |\mathcal{N}|} \quad \implies \quad N_{u,v,\ell}^{\mathrm{disp}} = \left[\mathbf{N}_\ell^{\mathrm{disp}}\right]_{u,v}. \tag{10}$$

Let $S_\ell^{\mathrm{disp}} = (H_\ell b + s_{\mathrm{prob}})$ and $S_\ell^{\mathrm{comb}} = (H_\ell b)$ be per-token bytes. The byte-volume matrices are

$$\mathbf{B}_\ell^{\mathrm{disp}} = S_\ell^{\mathrm{disp}} \mathbf{N}_\ell^{\mathrm{disp}}, \qquad \mathbf{B}_\ell^{\mathrm{comb}} = S_\ell^{\mathrm{comb}} \mathbf{N}_\ell^{\mathrm{disp}}.$$

**Direction-specific link model.**   From profiling (Alg. 1), let

$$\boldsymbol{\alpha}^{\mathrm{d}}, \ \boldsymbol{\beta}^{\mathrm{d}}, \ \boldsymbol{\alpha}^{\mathrm{c}}, \ \boldsymbol{\beta}^{\mathrm{c}} \in \mathbb{R}^{|\mathcal{N}| \times |\mathcal{N}|}$$

collect the dispatch/combine intercepts and per-byte slopes (Hadamard product $\odot$ below). A dispatch $u \to v$ plus a combine $v \to u$ yields the traffic-weighted time proxy

$$\mathbf{C}_\ell^{\mathrm{tw}} = \boldsymbol{\alpha}^{\mathrm{d}} + \boldsymbol{\beta}^{\mathrm{d}} \odot \mathbf{B}_\ell^{\mathrm{disp}} + \left(\boldsymbol{\alpha}^{\mathrm{c}}\right)^\top + \left(\boldsymbol{\beta}^{\mathrm{c}}\right)^\top \odot \mathbf{B}_\ell^{\mathrm{comb}}. \tag{11}$$

**Row-wise z-score.**   Let $\mathbf{1} \in \mathbb{R}^{|\mathcal{N}|}$ be the all-ones vector. Define row means and standard deviations:

$$\boldsymbol{\mu}_\ell = \tfrac{1}{|\mathcal{N}|} \mathbf{C}_\ell^{\mathrm{tw}} \mathbf{1} \in \mathbb{R}^{|\mathcal{N}| \times 1}, \qquad \boldsymbol{\sigma}_\ell = \mathrm{std}_{\mathrm{row}}\left(\mathbf{C}_\ell^{\mathrm{tw}}\right) \in \mathbb{R}^{|\mathcal{N}| \times 1}.$$

Then the row-wise standardized matrix is

$$\widehat{\mathbf{C}}_\ell^{\mathrm{tw}} = \left(\mathbf{C}_\ell^{\mathrm{tw}} - \boldsymbol{\mu}_\ell \mathbf{1}^\top\right) \oslash \left(\boldsymbol{\sigma}_\ell \mathbf{1}^\top + \varepsilon\right), \tag{12}$$

so each row is zero-mean and unit-variance across destinations $v$.

**Topology-aware gate bias.**   Precompute a gate-bias table and inject it into logits:

$$\mathbf{G}^{(\ell)} := -\lambda_{\mathrm{cost}} \widehat{\mathbf{C}}_\ell^{\mathrm{tw}} \mathbf{M}_\ell^\top \in \mathbb{R}^{|\mathcal{N}| \times E} \quad \Rightarrow \quad \mathbf{Z}'^{(\ell)} = \mathbf{Z}^{(\ell)} + \mathbf{S}\mathbf{G}^{(\ell)}. \tag{13}$$

Here $\lambda_{\mathrm{cost}}$ controls the strength of the bias; the z-score makes it dimensionless and comparable across sources.

**Multi-hop & asymmetry:**   Prefer end-to-end, per-direction fitted $(\alpha_{uv}, \beta_{uv})$; if only per-hop parameters are available, aggregate via store-and-forward (sum) or cut-through/pipelined (sum on $\alpha$, max on $\beta$).

**Dynamic adaptation:**   A lightweight controller can tune $\lambda_{\mathrm{cost}}$ from observed vs. predicted A2A time and sparsely micro-probe predicted bottlenecks to refresh a few $(\alpha, \beta)$; overhead is negligible and off the training fast path.

**Complexity.**   Online cost reduces to a single gather-and-add $\mathbf{S}\mathbf{G}^{(\ell)}$ per layer. Stacking $\{\mathbf{G}^{(\ell)}\}_{\ell=1}^L$ yields, for each source endpoint $u$, a tiny cache $\mathbf{G}_u \in \mathbb{R}^{L \times E}$ (8 KiB in FP16 for $L{=}64$, $E{=}64$).

## 4 EVALUATION

### 4.1 EXPERIMENT SETUP

**Implementation**   We implemented a prototype of **CAMOE** based on Megatron-LM, consisting of approximately 1.5k lines of Python and 0.5k lines of C++. Given that the `alltoall` communication is synchronous and blocking, it is impossible to probe the status of all communication links within a single regular `alltoall` operation. Therefore, we introduced a new token dispatcher to generate sufficient communication data, ensuring high accuracy in cost model fitting.

By reusing the parallel state abstraction provided by Megatron-core, we can invoke built-in `alltoall` primitives without modifying the core framework code. This approach allows us to accurately record timing information for various computational operations within the framework, such as statistical aggregation, reordering, and sorting. Compared to real inference scenarios, our system provides a more precise measurement of link status, as it is free from routing uncertainties.

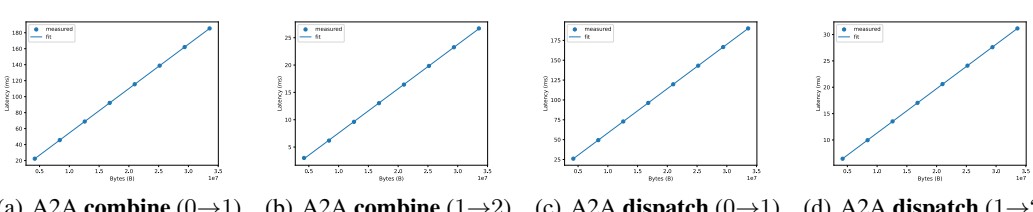

(a) A2A **combine** (0→1)   (b) A2A **combine** (1→2)   (c) A2A **dispatch** (0→1)   (d) A2A **dispatch** (1→2)

Figure 6: Verification of $\alpha$–$\beta$ fits for four All-to-All cases. Each panel shows measured latencies across message sizes with the fitted model overlaid.

**Hardware**   We conducted our experiments using four NVIDIA ADA6000 GPUs (48GB each), dual Intel(R) Xeon(R) Gold 6544Y CPUs, and 2TB of RAM.

**Environment Setup**   We captured latency and bandwidth data from actual nodes in real cloud service scenarios and utilized Docker to configure a distributed environment that traverses network stacks, emulating real public cloud environments. Detailed setup instructions can be found in the appendixA.3.

### 4.2   EVALUATION BENCHMARKS

We evaluate with the `lm-evaluation-harness` (Gao et al., 2024) on nine standard downstream tasks with zero/few-shot: BoolQ (0-shot, Accuracy)(Clark et al., 2019), RTE (0-shot, Accuracy)(Wang et al., 2018), OBQA (0-shot, Accuracy with length normalization)(Mihaylov et al., 2018), PIQA (0-shot, Accuracy with length normalization)(Bisk et al., 2019), MMLU (5-shot, Accuracy)(Hendrycks et al., 2021), WinoGrande (5-shot, Accuracy)(Sakaguchi et al., 2019), GSM8K (5-shot, Exact Match)(Cobbe et al., 2021), HellaSwag (10-shot, Accuracy with length normalization)(Zellers et al., 2019), and ARC-C (25-shot, Accuracy with length normalization)(Clark et al., 2018).

### 4.3   SIMULATOR FIDELITY EVALUATION

We verify the $\alpha$–$\beta$ model using a single micro-benchmark sweep in which the number of tokens sent per All-to-All ranges from 128 to 1024 in steps of 128. For each case we fit $t = \alpha + \beta n$ (with $t$ in milliseconds and $n$ in bytes), obtaining: **dispatch** $0 \to 1$ $\hat{\alpha} = 2.5480\,\text{ms}$, $\hat{\beta} = 5.5823 \times 10^{-6}\,\text{ms/byte}$; **dispatch** $1 \to 2$ $\hat{\alpha} = 2.9142\,\text{ms}$, $\hat{\beta} = 8.4092 \times 10^{-7}\,\text{ms/byte}$; **combine** $0 \to 1$ $\hat{\alpha} = 0.9744\,\text{ms}$, $\hat{\beta} = 5.5532 \times 10^{-6}\,\text{ms/byte}$; **combine** $1 \to 2$ $\hat{\alpha} = 0.9454\,\text{ms}$, $\hat{\beta} = 8.0976 \times 10^{-7}\,\text{ms/byte}$. As shown in Fig. 6, the fitted curves closely track the measured latencies across the entire size range for both dispatch and combine phases.

### 4.4   MAIN RESULTS

We evaluate the cost-aware routing coefficient $\lambda_{\text{cost}}$ on **Qwen3-30B-A3B** by sweeping $\{0, 0.05, 0.10, 0.15, 0.20, 0.25\}$ and measuring (i) mean forward All-to-All latency per MoE layer (ms) and (ii) accuracy on nine downstream tasks (BoolQ, OpenBookQA (norm), PIQA (norm), RTE, MMLU, WinoGrande, ARC-Challenge, HellaSwag, GSM8K). $\lambda_{\text{cost}}$ scales a $z$-scored, $\alpha$–$\beta$-derived per-(source node, expert) communication-cost bias added to gating logits; larger values steer tokens toward lower-cost experts without changing model parameters or FLOPs. Results are summarized in Table 1.

**Latency.**   All-to-All latency decreases monotonically with $\lambda_{\text{cost}}$: from **177.56 ms** at 0 to **149.50 ms** at 0.25 ($-28.06\,\text{ms}$; $-\mathbf{15.8}\%$). Intermediate settings yield smooth improvements: 171.49 ms ($-3.4\%$) at 0.05, 166.02 ms ($-6.5\%$) at 0.10, 160.24 ms ($-9.8\%$) at 0.15, and 152.51 ms ($-14.1\%$) at 0.20. These per-layer savings compound across MoE layers (subject to overlap), directly targeting the communication bottleneck. **Tail latency (p95)** shows a similarly monotonic drop: from **235.51 ms** at 0 to **190.57 ms** at 0.25 ($-44.94\,\text{ms}$; $-\mathbf{19.1}\%$). Intermediate settings are 220.88 ms

| $\lambda_{\text{cost}}$ | All-to-All mean (ms) | All-to-All p95 (ms) | Downstream Tasks | | | | | | | | |
|---|---|---|---|---|---|---|---|---|---|---|---|
| | | | BoolQ | OpenBookQA | PIQA | RTE | MMLU | WinoGrande | ARC-Challenge | HellaSwag | GSM8K |
| 0.00 | 177.56 | 235.51 | 0.8856 | **0.4500** | **0.8047** | 0.8231 | **0.7960** | 0.6993 | **0.6980** | **0.7792** | **0.8984** |
| 0.05 | 171.49 | 220.88 | 0.8872 | 0.4420 | 0.8036 | 0.8303 | 0.7953 | **0.7088** | 0.6869 | 0.7789 | 0.8939 |
| 0.10 | 166.02 | 209.50 | **0.8890** | 0.4420 | 0.7911 | 0.8281 | 0.7945 | 0.6993 | 0.6928 | 0.7767 | 0.8817 |
| 0.15 | 160.24 | 208.43 | 0.8829 | 0.4420 | 0.7867 | 0.8195 | 0.7913 | 0.6985 | 0.6843 | 0.7751 | 0.8802 |
| 0.20 | 152.51 | 197.80 | 0.8795 | 0.4300 | 0.7709 | 0.8267 | 0.7893 | 0.6961 | 0.6877 | 0.7709 | 0.8855 |
| 0.25 | 149.50 | 190.57 | 0.8780 | 0.4280 | 0.7650 | **0.8375** | 0.7822 | 0.6977 | 0.6860 | 0.7640 | 0.8840 |

Table 1: Evaluating the cost-aware routing coefficient ($\lambda_{\text{cost}}$) on the Qwen3-30B-A3B MoE model. Column 2 reports *mean* forward All-to-All latency per MoE layer (ms; lower is better); Column 3 reports *p95* (95th percentile) forward All-to-All latency (ms). Remaining columns list accuracies on nine downstream tasks. **Best** per task is bolded. Larger $\lambda_{\text{cost}}$ biases routing toward lower-cost experts, reducing latency with generally minor accuracy movement.

($-6.2\%$) at 0.05, 209.50 ms ($-11.0\%$) at 0.10, 208.43 ms ($-11.5\%$) at 0.15, and 197.80 ms ($-16.0\%$) at 0.20, indicating that cost-aware routing reduces not only mean but also tail All-to-All latency, with larger relative gains in the tail.

**Accuracy.** Best scores are dispersed across $\lambda_{\text{cost}}$ (bold in the table), and most tasks remain near baseline. Notable *improvements* include *RTE* peaking at 0.25 (+0.0144 abs., 0.8231→0.8375) and *WinoGrande* at 0.05 (+0.0095 abs., 0.6993→0.7088); both changes are comparable to their reported $\pm$ intervals. The strongest *declines* at high $\lambda$ appear on *PIQA* (0.8047→0.7650, $-0.0397$), *MMLU* (0.7960→0.7822, $-0.0138$), *HellaSwag* (0.7792→0.7640, $-0.0152$), *GSM8K* (0.8984→0.8840, $-0.0144$), and *OpenBookQA* (0.4500→0.4280, $-0.0220$); several of these exceed the typical reported uncertainty bands (e.g., $\pm0.003$–0.013), indicating a real trade-off for those tasks. On average (unweighted across nine tasks), accuracy at 0.25 is $\sim$1.2 pp below baseline.

### 4.5 Expert Choice Analysis

With fixed expert placement and dataset, Table 2 shows a monotonic rise in global selectivity: `cv` increases from 0.3368 to 0.3742 (+11.1%) as $\lambda_{\text{cost}}$ grows, indicating tokens shift toward cheaper-link experts. Crucially, per-layer imbalance remains flat (`avg_layer_cv` $\approx$ 1.51), implying Top-$K$ gating and capacity limits preserve within-layer balance and avoid saturation. The routing distribution changes smoothly: $\text{KL}(\lambda : 0)$ rises from $5.6 \times 10^{-4}$ to $1.49 \times 10^{-2}$ (with `avg_layer_kl` $\leq 5.9 \times 10^{-2}$), which may slightly affect precision-sensitive

| $\lambda_{\text{cost}}$ | cv | avg_layer_cv | KL ↓ | avg_layer_kl ↓ |
|---|---|---|---|---|
| 0.0000 | 0.3368 | 1.5118 | 0.0000 | 0.0000 |
| 0.0500 | 0.3379 | 1.5122 | 0.0006 | 0.0098 |
| 0.1000 | 0.3437 | 1.5137 | 0.0023 | 0.0213 |
| 0.1500 | 0.3525 | 1.5132 | 0.0053 | 0.0321 |
| 0.2000 | 0.3622 | 1.5108 | 0.0097 | 0.0447 |
| 0.2500 | 0.3742 | 1.5079 | 0.0149 | 0.0588 |

Table 2: Expert selectivity & routing shift vs. $\lambda_{\text{cost}}$. Higher **cv** indicates stronger selectivity; **avg_layer_cv** is the mean per-layer CV. KLs are measured against the $\lambda$=0 baseline (nats).

tasks (e.g., PIQA, HellaSwag, GSM8K). A practical Pareto region is $\lambda_{\text{cost}} \in [0.15, 0.20]$, yielding `cv` gains of 4.7%–7.6% with KL at the $10^{-2}$ scale, steering traffic onto shorter paths and reducing All-to-All latency with minimal quality impact.

## 5 Conclusions

As Mixture-of-Experts (MoE) models scale by distributing experts across increasingly heterogeneous hardware, communication overhead, particularly tail latency from All-to-All operations, becomes the dominant inference bottleneck. This work introduces **CAMoE**, a framework that mitigates this by making routing cost-aware. We profile system topology to model communication latency and inject a lightweight, topology-aware bias into the gating function at inference time. CAMoE reduces mean and tail All-to-All latency by up to 15.8% and 19.1% on a 30B-parameter model, with minimal accuracy impact and no retraining.

## 6 LLM Usage

During the writing process of this paper, a large language model (LLM) was used only for minor text polishing and spell checking. The research design, experimental analysis, and conclusions were independently completed by the authors. The LLM was not used to generate data, code, or results.

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

# A APPENDIX

## A.1 ALGORITHM

---

**Algorithm 1** Two-Phase $\alpha$–$\beta$ Profiling

---

**Input:** endpoint count $N$, message-size set $\mathcal{M}$.
**Output:** $\{(\alpha_{uv}^p, \beta_{uv}^p)\}_{u \neq v, \, p \in \{m,d,c\}}$.
**Phase 1: Baseline for** $u \leftarrow 0$ **to** $N-1$ **do**
   **foreach** $v \in [0, N-1] \setminus \{u\}$ **do**
      **foreach** $p \in \{m, d, c\}$ **do**
         $\mathcal{D}_{uv}^{0,p} \leftarrow \emptyset$ **foreach** $M \in \mathcal{M}$ **do**
            $\mathcal{P} \leftarrow \text{ISOPATTERN}(u, v, M, p)$     // only $(u{\to}v)$ active in phase $p$
            $B \leftarrow \text{BYTES}(u, v, \mathcal{P}, p)$,   $t \leftarrow \text{TIME}(\mathcal{P})$  $\mathcal{D}_{uv}^{0,p} \leftarrow \mathcal{D}_{uv}^{0,p} \cup \{(B, t)\}$
         **end**
         $(\alpha_{uv}^{0,p}, \beta_{uv}^{0,p}) \leftarrow \text{FIT}(\mathcal{D}_{uv}^{0,p})$
      **end**
   **end**
**end**
**Phase 2: Congestion-aware for** $u \leftarrow 0$ **to** $N-1$ **do**
   **foreach** $M \in \mathcal{M}$ **do**
      $\mathcal{P} \leftarrow \text{ONETOMANY}(u, M)$  $v^\star \leftarrow \arg\max_{v \neq u} \alpha_{uv}^{0,d} + \beta_{uv}^{0,d} \cdot \text{BYTES}(u, v, \mathcal{P}, d)$  $t_d \leftarrow$
      $\text{TIME}(\mathcal{P})$  $\mathcal{D}_{uv^\star}^d \leftarrow \mathcal{D}_{uv^\star}^d \cup \{(\text{BYTES}(u, v^\star, \mathcal{P}, d), t_d)\}$
      // Use the symmetric choice for combine
      $v^\star \leftarrow \arg\max_{v \neq u} \alpha_{vu}^{0,c} + \beta_{vu}^{0,c} \cdot \text{BYTES}(v, u, \mathcal{P}, c)$  $t_c \leftarrow \text{TIME}(\mathcal{P})$  $\mathcal{D}_{v^\star u}^c \leftarrow \mathcal{D}_{v^\star u}^c \cup$
      $\{(\text{BYTES}(v^\star, u, \mathcal{P}, c), t_c)\}$
   **end**
**end**
**forall** $u \neq v$ **do**
   $(\alpha_{uv}^d, \beta_{uv}^d) \leftarrow \text{FIT}(\mathcal{D}_{uv}^d)$;   $(\alpha_{uv}^c, \beta_{uv}^c) \leftarrow \text{FIT}(\mathcal{D}_{uv}^c)$;   $(\alpha_{uv}^m, \beta_{uv}^m) \leftarrow (\alpha_{uv}^{0,m}, \beta_{uv}^{0,m})$.
**end**

---

## A.2 RELATED WORK

**Mixture-of-Experts Models.** Early MoE work introduced sparse expert routing with auxiliary balancing losses(Shazeer et al., 2017; Lepikhin et al., 2020; Fedus et al., 2022). Later variants explored adaptive expert selection and lightweight modifications such as gating residuals or zero-compute experts (Yang et al., 2024; Li et al., 2023; Jin et al., 2024). These efforts mainly target model quality and theoretical efficiency, but generally assume homogeneous all-to-all communication and measure gains in FLOPs rather than real deployment latency. Recent large-scale systems such as DeepSeek-V3 (DeepSeek-AI et al., 2025) and Qwen3-MoE (Yang et al., 2025) demonstrate the practicality of scaling sparse activation to hundreds of billions of parameters, highlighting the need to also optimize system-level efficiency.

**System optimization** A complementary line of work addresses the performance bottlenecks of distributed MoE. FastMoE (He et al., 2021) and Tutel (Hwang et al., 2023) improve scalability through expert placement and overlapping communication with computation. Other frameworks introduce padding-free dispatch, hybrid parallelism, or hierarchical deduplication (Yuan et al., 2025; Lin et al., 2025). Most relevant to our approach, NetMoE (Liu et al., 2025) and MoETuner (Go & Mahajan, 2025) formulate topology-aware token routing and expert placement strategies, explicitly modeling communication cost. However, these systems still assume relatively stable or offline-optimized topologies. Our work differs by profiling heterogeneous links at inference time and incorporating a lightweight, cost-aware gating bias without retraining.

## A.3 Environment Settings

**Node1**

```
tc qdisc add dev eth0 root handle 1: prio bands 3
tc qdisc add dev eth0 parent 1:1 handle 10: netem delay 0.65ms
tc qdisc add dev eth0 parent 10: handle 11: tbf rate 1500mbit burst 2mbit latency 50ms
tc filter add dev eth0 protocol ip parent 1: prio 1 u32 match ip dst 172.17.0.4/32 flowid 1:1
tc qdisc add dev eth0 parent 1:2 handle 20: netem delay 0.56ms
tc qdisc add dev eth0 parent 20: handle 21: tbf rate 1500mbit burst 2mbit latency 50ms
tc filter add dev eth0 protocol ip parent 1: prio 1 u32 match ip dst 172.17.0.5/32 flowid 1:2
tc qdisc add dev eth0 parent 1:3 handle 30: netem delay 0.55ms
tc qdisc add dev eth0 parent 30: handle 31: tbf rate 1500mbit burst 2mbit latency 50ms
tc filter add dev eth0 protocol ip parent 1: prio 1 u32 match ip dst 172.17.0.6/32 flowid 1:3
```

**Node2**

```
tc qdisc add dev eth0 root handle 1: prio bands 3
tc qdisc add dev eth0 parent 1:1 handle 10: netem delay 0.645ms
tc qdisc add dev eth0 parent 10: handle 11: tbf rate 1500mbit burst 2mbit latency 50ms
tc filter add dev eth0 protocol ip parent 1: prio 1 u32 match ip dst 172.17.0.3/32 flowid 1:1
tc qdisc add dev eth0 parent 1:2 handle 20: netem delay 0.63ms
tc qdisc add dev eth0 parent 20: handle 21: tbf rate 10000mbit burst 10mbit latency 50ms
tc filter add dev eth0 protocol ip parent 1: prio 1 u32 match ip dst 172.17.0.5/32 flowid 1:2
tc qdisc add dev eth0 parent 1:3 handle 30: netem delay 0.65ms
tc qdisc add dev eth0 parent 30: handle 31: tbf rate 10000mbit burst 10mbit latency 50ms
tc filter add dev eth0 protocol ip parent 1: prio 1 u32 match ip dst 172.17.0.6/32 flowid 1:3
```

**Node3**

```
tc qdisc add dev eth0 root handle 1: prio bands 3
tc qdisc add dev eth0 parent 1:1 handle 10: netem delay 0.56ms
tc qdisc add dev eth0 parent 10: handle 11: tbf rate 1500mbit burst 2mbit latency 50ms
tc filter add dev eth0 protocol ip parent 1: prio 1 u32 match ip dst 172.17.0.3/32 flowid 1:1
tc qdisc add dev eth0 parent 1:2 handle 20: netem delay 0.63ms
tc qdisc add dev eth0 parent 20: handle 21: tbf rate 10000mbit burst 10mbit latency 50ms
tc filter add dev eth0 protocol ip parent 1: prio 1 u32 match ip dst 172.17.0.4/32 flowid 1:2
tc qdisc add dev eth0 parent 1:3 handle 30: netem delay 70us
tc qdisc add dev eth0 parent 30: handle 31: tbf rate 10000mbit burst 10mbit latency 50ms
tc filter add dev eth0 protocol ip parent 1: prio 1 u32 match ip dst 172.17.0.6/32 flowid 1:3
```

**Node4**

```
tc qdisc add dev eth0 root handle 1: prio bands 3
tc qdisc add dev eth0 parent 1:1 handle 10: netem delay 0.565ms
tc qdisc add dev eth0 parent 10: handle 11: tbf rate 1500mbit burst 2mbit latency 50ms
tc filter add dev eth0 protocol ip parent 1: prio 1 u32 match ip dst 172.17.0.3/32 flowid 1:1
tc qdisc add dev eth0 parent 1:2 handle 20: netem delay 0.65ms
tc qdisc add dev eth0 parent 20: handle 21: tbf rate 10000mbit burst 10mbit latency 50ms
tc filter add dev eth0 protocol ip parent 1: prio 1 u32 match ip dst 172.17.0.4/32 flowid 1:2
tc qdisc add dev eth0 parent 1:3 handle 30: netem delay 65us
tc qdisc add dev eth0 parent 30: handle 31: tbf rate 10000mbit burst 10mbit latency 50ms
tc filter add dev eth0 protocol ip parent 1: prio 1 u32 match ip dst 172.17.0.5/32 flowid 1:3
```

## A.4 Metrics

Let $\mathcal{E}$ be the expert set with $E = |\mathcal{E}|$ and $\mathcal{L}$ be the set of MoE layers with $L = |\mathcal{L}|$. Let $n_{\ell,e}$ denote the (post-drop) token count processed by expert $e \in \mathcal{E}$ at layer $\ell \in \mathcal{L}$.

**Global coefficient of variation (cv).** Aggregate expert loads across layers: $n_e = \sum_{\ell \in \mathcal{L}} n_{\ell,e}$. Define

$$\bar{n} = \frac{1}{E} \sum_{e \in \mathcal{E}} n_e, \quad s = \sqrt{\frac{1}{E} \sum_{e \in \mathcal{E}} (n_e - \bar{n})^2}, \quad \mathrm{cv} = \frac{s}{\bar{n}}. \tag{14}$$

A larger cv indicates stronger global selectivity (more concentration on a subset of experts).

**Per-layer coefficient of variation (avg_layer_cv).** For each layer $\ell$, compute

$$\bar{n}_\ell = \frac{1}{E} \sum_{e \in \mathcal{E}} n_{\ell,e}, \qquad s_\ell = \sqrt{\frac{1}{E} \sum_{e \in \mathcal{E}} (n_{\ell,e} - \bar{n}_\ell)^2}, \qquad \mathrm{cv}_\ell = \frac{s_\ell}{\bar{n}_\ell}. \tag{15}$$

Then

$$\mathrm{avg\_layer\_cv} = \frac{1}{L} \sum_{\ell \in \mathcal{L}} \mathrm{cv}_\ell. \tag{16}$$

**Routing shift KL (KL).** We compare a baseline run (*base*) and a comparison run (*cmp*) via layer-aggregated expert-load histograms. Let $c_e^{\mathrm{base}} = \sum_\ell n_{\ell,e}^{\mathrm{base}}$ and $c_e^{\mathrm{cmp}} = \sum_\ell n_{\ell,e}^{\mathrm{cmp}}$ be global expert counts. Convert them to probability distributions with $\varepsilon$-smoothing:

$$P_e = \frac{c_e^{\mathrm{base}} + \varepsilon}{\sum_{j \in \mathcal{E}} c_j^{\mathrm{base}} + \varepsilon E}, \qquad Q_e = \frac{c_e^{\mathrm{cmp}} + \varepsilon}{\sum_{j \in \mathcal{E}} c_j^{\mathrm{cmp}} + \varepsilon E}. \tag{17}$$

The global KL divergence (in nats) is

$$\mathrm{KL} = D_{\mathrm{KL}}(P \| Q) = \sum_{e \in \mathcal{E}} P_e \log \frac{P_e}{Q_e}. \tag{18}$$

**Average per-layer KL (avg_layer_kl).** Form layer-wise distributions by smoothing and normalizing counts per layer:

$$P_e^{(\ell)} = \frac{n_{\ell,e}^{\mathrm{base}} + \varepsilon}{\sum_{j \in \mathcal{E}} n_{\ell,j}^{\mathrm{base}} + \varepsilon E}, \qquad Q_e^{(\ell)} = \frac{n_{\ell,e}^{\mathrm{cmp}} + \varepsilon}{\sum_{j \in \mathcal{E}} n_{\ell,j}^{\mathrm{cmp}} + \varepsilon E}. \tag{19}$$

Compute per-layer KLs $D_{\mathrm{KL}}\big(P^{(\ell)} \| Q^{(\ell)}\big) = \sum_e P_e^{(\ell)} \log \frac{P_e^{(\ell)}}{Q_e^{(\ell)}}$ and average:

$$\mathrm{avg\_layer\_kl} = \frac{1}{L} \sum_{\ell \in \mathcal{L}} D_{\mathrm{KL}}\big(P^{(\ell)} \| Q^{(\ell)}\big). \tag{20}$$