# OpenReview forum: "CAMoE: Cost-Aware Communication Optimization for Mixture-of-Experts Inference"
_ICLR.cc/2026/Conference — ICLR 2026 Conference Withdrawn Submission_

### Official Review · Reviewer_w3MD · 2025-10-16

**Soundness:** 3
**Presentation:** 1
**Contribution:** 2
**Rating:** 2
**Confidence:** 5

**Summary:**

The paper addresses inference-time latency in Mixture-of-Experts models that employ expert parallelism (each expert MLP slice is hosted on a separate device). This parallelism strategy introduces significant communication overhead: each MoE layer requires an All-to-All dispatch (to send token embeddings to their selected experts) and an All-to-All combine (to gather processed outputs back). The dispatch phase, in particular, contributes to high latency. Since the total all-to-all latency depends on the slowest pair, specifically To mitigate this, the authors propose a profiler-based approach that measures per-link communication characteristics (, ) across GPUs and incorporates these estimates into the routing bias—so that the gate function implicitly prefers experts located on faster communication paths. In essence, the router becomes communication-aware, balancing accuracy against network cost. Experiments on the Qwen-30B MoE model show modest latency reductions without noticeable accuracy degradation.

Despite that I very much like the idea of making router communication aware, I think the paper needs more work (positioning, writing, experiments and baselines) to be accepted. The paper is very difficult to follow and the experiment section is very limited (see weakness). I also have some question regarding the evaluation setup (see questions).

I would encourage authors to work in this direction, however I think the paper is not ready yet to be published.

**Strengths:**

The problem the paper addresses is real and well-documented in prior literature. I also find the core idea of making the routing mechanism communication-aware interesting and conceptually valuable.

**Weaknesses:**

The overall positioning and writing style of the paper raise several disagreements with how the authors frame and present their findings:

For example, the authors state (Section 2):

> However, a critical and often-overlooked challenge in the real-world deployment of modern multiGPU servers is the extreme heterogeneity of intra-node communication. While it is commonly assumed that inter-GPU communication within the same server is homogeneous and efficient, our empirical measurements reveal a starkly different reality.

I don't think this is a correct statement, people who operate large-scale GPU clusters do not actually assume that intra-node communication is homogeneous, unless the system indeed uses a uniform interconnect. Calling this an “overlooked challenge” is therefore misleading. If the authors believe otherwise, they should support this claim with references or survey evidence showing that this misconception exists in practice.

> As shown in Figure3, depending on the physical interconnect topology between a pair of GPUs, their pointto-point write latency can differ by orders of magnitude.

This is not a surprising empirical discovery, it reproduces well-known and well-documented hardware behavior. A simple reference to NCCL documentation or NVIDIA’s public materials would be sufficient. The same comment applies to Figure 5.

Subsection 2.2 spends considerable space re-stating a well-known hardware fact: that latency depends on the type of interconnect and that All-to-All communication is limited by the slowest peer. This discussion feels redundant—particularly since the main Algorithm 1 is already placed in the Appendix. I would recommend that the authors either remove this paragraph entirely or condense it significantly. Given that the proposed algorithm itself does not explicitly differentiate between intra- and inter-node communication, the space would be better used to provide a clearer and more detailed description of the proposed mathematical communication model.

2. Lack of baselines and experiments

After reading the paper, I understand that the main contribution is its applicability to heterogeneous hardware setups. In the Related Work section (Appendix), the authors state:
>However, these systems still assume relatively stable or offline-optimized topologies. Our work differs by profiling heterogeneous links at inference time and incorporating a lightweight, cost-aware gating bias without retraining.

However, I found multiple prior works that also optimize for heterogeneous clusters, which are not mentioned or compared against in this paper. For example, https://arxiv.org/abs/2503.04398 (see Subsection 5.1(B)) presents a relevant approach. I would strongly recommend that the authors include comparisons to such works or clearly justify why they are not directly relevant.

In the introduction, the authors emphasize the importance of handling heterogeneous intra-node communication, so I expected the experiments to demonstrate that the proposed profiler is effective under a range of connectivity types. However, in Appendix A.3 the provided configuration appears to focus on heterogeneous inter-node communication instead. It is not clear to me why this particular setup was chosen.

More specifically, the reported latencies are confusing: for example, the latency from node 3 to node 4 is given as ~70 microseconds, while the latency from node 1 to node 2 is given as ~65 milliseconds. This is a difference of roughly three orders of magnitude, and the paper never explains why this disparity exists, whether it is intentional, or whether it reflects real hardware. I think the paper would benefit from a clearer description of the hardware topology, how these latency numbers were obtained, and why these particular node pairs were chosen.

Related to that, it would also strengthen the work to vary the communication latency across nodes in a controlled way. Right now, having one link that is ~100× faster than the others feels a bit ad hoc; if this is actually representative of real clusters, a citation or justification would make that much more convincing.

---
Small corrections:

* line 33, 223: \cite instead of \citep?
* line 205: space is missing
* Figure 4 takes considerable amount of space, but mentioned only one time in the paper and looks redundant

**Questions:**

1. > By reusing the parallel state abstraction provided by Megatron-core, we can invoke built-in alltoall primitives without modifying the core framework code.

Could you elaborate on what was actually changed or configured?

2. It is not clear how much overhead running the profiler contributed to the overall latency. Can authors provide details?
3. No warmups/repetitions:
> We estimate (α_{uv}, β_{uv}) with no warmups or repeated trials.

Did the authors observe differences when warmups or repetitions were included?

---

### Official Review · Reviewer_y8qw · 2025-10-31

**Soundness:** 2
**Presentation:** 3
**Contribution:** 2
**Rating:** 4
**Confidence:** 5

**Summary:**

This paper investigates a peculiar but crucial topic, i.e, how to implement efficient all-to-all communication considering physical configurations during MoE model inference. The communication bandwidth across different GPUs or nodes is typically the main bottleneck in the serving of MoE-based LLMs with expert parallelism, and CAMoE tries to alleviate this issue by injecting a lightweight, topology-aware bias into the router’s gating decisions during inference, steering tokens away from high-cost communication paths without retraining. The key contributions of this paper include:
1. CAMoE Method: A drop-in, inference-time routing bias that incorporates communication cost into expert selection, reducing both mean and tail latency of All-to-All communication.
2. System Modeling and Profiling: A two-phase profiler and $\alpha-\beta$ communication model to estimate per-link latency and congestion, enabling accurate cost-aware routing.
3. Practical Integration: Seamlessly integrates with existing MoE systems (e.g., Megatron-LM), supporting various parallelism strategies without modifying model parameters or training.
4. Empirical Results: On the Qwen3-30B-A3B model, CAMoE reduces mean All-to-All latency by up to 15.8% and p95 latency by up to 19.1%, with minimal impact on downstream task accuracy.
5. Open-Source Toolkit: Provides a lightweight profiling and simulation toolkit for cost-aware MoE routing.

**Strengths:**

This paper develops a cost-aware method designed to reduce tail latency during model inference, which does not affect model accuracy on downstream tasks and effectively reduces all-to-all communication time during inference.
1. The topic of your work is quite interesting and practical, targeting a bottleneck problem in LLM serving.
2. CAMoE is training-free and does not significantly harm the model performance on downstream tasks.
3. The core design of CAMoE is simple and effective, which can be easily integrated into Megatron or other frameworks.

**Weaknesses:**

There are some key drawbacks in this submission, including:
1. Figure 1 may not be accurate and precise because
    - You did not provide the detailed configuration of this profiling figure. Since this paper is targeted at inference, you should state whether this figure is derived from the prefilling or decoding stage, because in the prefilling stage, the communication burden of the MoE layer can be significantly higher than in the decoding stage.
    - You should specify (1) which model you are examining, (2) which stage you are testing, (3) the batch size, and (4) the context length.
2. In Figure 1, the all-to-all dispatch time is significantly higher than the all-to-all combine. However, this can be confusing, as the computation, memory movement, and communication patterns of these two operations are symmetric, and the computation burden here is very small, allowing for well-parallelized execution with specialized CUDA (in Tutel) or Triton (in MegaBlocks) kernels.
3. You forgot to cite some papers with a very similar topic or method, including Auxiliary-loss free load balancing (DeepSeekAI), Occult (ICML2025), and MegaBlocks (MLSys 2023).
4. Some of the concepts in this paper are not clarified, e.g, row-wise z-score and direction-specific $\alpha-\beta$ model. The explanation of them should also be provided.
5. More MoE-based LLMs should be investigated because they typically adopt different MoE strategies. For example, Mixtral-8x7B uses heavy individual experts and top-2 routing among 8 total experts. DeepSeek-V2-Lite and Moonlight-16B-A3B utilize lightweight individual experts and employ top-6 routing among 64 total experts, with shared experts also introduced. These models have different MoE patterns, and the inference strategies are slightly different from the Qwen3-30B-A3B model you used in this paper.
6. Just a reminder: Megatron is an LLM training framework. In industrial applications such as LLM serving, it is vLLM and SGLang that are typically preferred as the inference engine. But considering vLLM or SGLang are not as customizable as Megatron, it is also reasonable to use them in your paper.

**Questions:**

You should distinguish between the prefilling and decoding stages in the efficiency evaluation experiments, along with other detailed configurations on workload, such as batch size and context length.

---

### Official Review · Reviewer_RLFY · 2025-10-31

**Soundness:** 3
**Presentation:** 2
**Contribution:** 2
**Rating:** 4
**Confidence:** 2

**Summary:**

This paper introduces CAMoE, a training-free framework designed to mitigate the All-to-All communication bottleneck in large, multi-device MoE model inference. The core problem is that standard MoE routers are oblivious to the underlying hardware topology, often routing tokens over slow, high-latency network paths, which creates performance stragglers and increases tail latency. The proposed method addresses this by first profiling the communication costs between devices to create a cost model. It then injects a simple, cost-aware bias directly into the router's logits at inference time, discouraging the selection of experts located on high-cost paths. The authors evaluate this on the Qwen3-30B-A3B MoE model, showing that it can reduce All-to-All latency by up to 19.1% with only a minor impact on downstream task accuracy.

**Strengths:**

- The paper addresses a critical and practical bottleneck in the deployment of very large MoE models.
- The authors provide a clear and compelling motivation for their work, including an excellent empirical analysis of the heterogeneity of communication costs in modern multi-GPU servers.
- The proposed method is simple to implement at inference time, boiling down to a single gather-and-add operation before the top-k selection.

**Weaknesses:**

**Critically Limited Experimental Scope:** The paper's biggest weakness is its empirical evaluation. The method is evaluated on only a single MoE model (Qwen3-30B-A3B) and at a small scale (4 GPUs). Furthermore, there are no comparisons to any alternative methods. The baseline is simply CAMoE with the cost parameter set to zero. This is insufficient to demonstrate the general applicability or competitiveness of the approach. The authors must expand their evaluation. This should include:
1) At least one other MoE model to show the method is not specific to one architecture.
2) Comparisons to relevant baselines, such as other topology-aware routing or expert placement methods (e.g., NetMoE, MoETuner), to properly contextualize the performance. Even a comparison to a simple, non-trained heuristic (e.g., prefer experts on the same node) would be informative.
3) Evaluation at a larger scale (e.g., 8 or 16 GPUs across multiple nodes) to validate the approach in a more realistic deployment scenario.


**Lack of Clarity and Readability:** The paper is not easy to follow. The overall pipeline connecting the profiler, simulator, and the final inference-time biasing is not explained cohesively. In addition, some of the terms are not defined in the main text and moved to Appendix without a reference in the main sections. E.g., cv in Section 4.5, which is a Coefficient of Variation, is mentioned and described only in Appendix A.4 without a link in the text. It would be better if it the main text would refer a reader to the corresponding section.


In its current form, the paper presents an interesting proof-of-concept but lacks the rigorous and comprehensive evaluation required for ICLR. It would need to significantly expand its experimental evaluation to be a reasonable candidate.

**Questions:**

See above

---

### Official Review · Reviewer_mYre · 2025-11-02

**Soundness:** 2
**Presentation:** 2
**Contribution:** 2
**Rating:** 4
**Confidence:** 3

**Summary:**

CAMoE is a drop-in router-side tweak that aims squarely at tail latency, which dominates MoE inference and is bottlenecked by the slowest receiver. The aim is to cut MoE All-to-All communication latency by steering the router’s top-k expert choices toward lower-cost links. Concretely:

1. A lightweight two-phase profiler measures per-direction α–β link parameters over heterogeneous paths.

2. Using measured (α,β), the authors build a traffic-weighted time proxy for each source-endpoint - expert-endpoint pair and row-wise z-score it, producing a topology-aware bias that is added to the router logits before normal top-k selection.

3. The communication model and All-to-All timing are formalized, and the α–β fits are validated against microbenchmarks.

**Strengths:**

1. The paper provides a training-free generalizable method; no changes to collectives or retraining.

2. The detailed method is lightweight profiling and direction-specific α–β modeling with a neat two-phase procedure. The code integration is lightweight, with minimal code change (about 1.5k lines of Python code).

3. The results demonstrate clear monotonic latency reductions and good performance.

**Weaknesses:**

1. Experiments only run on 4× Ada 6000 with emulated network shaping. The validity of production environments is not demonstrated.

2.  No full E2E latency/throughput vs. batch/seq-len or cost-per-token is collected. Lack baseline with placement or topology-aware systems (e.g., NetMoE ICLR'25) and a simpler heuristics-based baseline.

3. Some tasks' performance is degraded in Table 1. The author should provide detailed reasons and analysis.

4. It's not clear how to online adapt \lambda_{\text{cost}} and refreshing (α,β).

5. Abstract references a SystemC-based simulator, but the main body of paper emphasizes Megatron-LM integration and α–β validation. The clarify the simulator’s role.

**Questions:**

1. How do improvements in per-layer A2A translate to full-model latency/throughput across realistic batch sizes and sequence lengths? Can you report tokens/s and time-to-first-token?
2. How does CAMoE perform on multi-node, NVSwitch/NVLink clusters with dozens to hundreds of GPUs and real spine/leaf congestion? Any results on p99?
3. Please compare against (a) placement-aware methods (Tutel, NetMoE, MoETuner) and (b) simpler biases (e.g., hop count / static bandwidth map) to isolate where α–β modeling adds value.   For tasks that regress at high \lambda_{\text{cost}} (PIQA, HellaSwag, GSM8K), do you observe changes in expert utilization/specialization (e.g., entropy of expert assignments, cross-layer affinity)? Can a per-layer \lambda_{\text{cost}} mitigate this?
4. What is the overhead and stability of an online controller that tunes \lambda_{\text{cost}} and refreshes a subset of (α,β) under drifting congestion? Any risk of oscillation or thrashing? How does the bias interact with capacity factors, token-drop, top-k choices (k=1 vs 2), and expert parallel group size?
5. Simulator vs. measurements: The abstract mentions a SystemC-based simulator; how is it used relative to the Megatron-based profiling/validation reported in Section 4? Can you reconcile and release both? What is the runtime overhead of the gather-and-add bias table lookup and the one-shot accounting pass? Can the profiler run in the background without disturbing SLA?

---

### Note · Authors · 2025-11-17

**Comment:**

Thank you for all reviewers‘ comments

**Withdrawal Confirmation:**

I have read and agree with the venue's withdrawal policy on behalf of myself and my co-authors.